# A unified form of low-energy nodal electronic interactions in hole-doped cuprate superconductors

T.J. Reber[1,5]*, X. Zhou [1]*, N.C. Plumb [1,6], S. Parham[1], J.A. Waugh[1], Y. Cao[1], Z. Sun[1,7], H. Li[1], Q. Wang[1], J.S. Wen [2], Z.J. Xu[2], G. Gu[2], Y. Yoshida[3], H. Eisaki[3], G.B. Arnold[1] & D.S. Dessau[1,4]*

Using angle resolved photoemission spectroscopy measurements of $Bi_2Sr_2CaCu_2O_{8+\delta}$ over a wide range of doping levels, we present a universal form for the non-Fermi liquid electronic interactions in the nodal direction in the exotic normal state phase. It is described by a continuously varying power law exponent versus energy and temperature (hence named a Power Law Liquid or PLL), which with doping varies smoothly from a quadratic Fermi Liquid in the overdoped regime, to a linear Marginal Fermi Liquid at optimal doping, to a non-quasiparticle non-Fermi Liquid in the underdoped regime. The coupling strength is essentially constant across all regimes and is consistent with Planckian dissipation. Using the extracted PLL parameters we reproduce the experimental optics and resistivity over a wide range of doping and normal-state temperature values, including the $T^*$ pseudogap temperature scale observed in the resistivity curves. This breaks the direct link to the pseudogapping of anti-nodal spectral weight observed at similar temperature scales and gives an alternative direction for searches of the microscopic mechanism.

[1] Department of Physics, University of Colorado, Boulder, CO 80309-0390, USA. [2] Condensed Matter Physics and Materials Science Department, Brookhaven National Labs, Upton, NY 11973, USA. [3] AIST Tsukuba Central 2, 1-1-1 Umezono, Tsukuba, Ibaraki 3058568, Japan. [4] Center for Experiments on Quantum Materials, University of Colorado, Boulder, CO 80309-0390, USA. [5] Present address: Department of Chemistry, University of Georgia, Athens, GA 30602, USA. [6] Present address: Swiss Light Source, Paul Scherrer Institut, CH-5232 Villigen PSI, Switzerland. [7] Present address: University of Science and Technology of China, Hefei, China. *email: ted.reber@gmail.com; xiaoqing.zhou@colorado.edu; dessau@colorado.edu

The non-superconducting state of the cuprates is debatably more intriguing and unusual than the high temperature superconducting state itself. One of the most iconic aspects, and a key to the electronic interactions out of which the superconducting state is borne, is the non-Fermi liquid resistivity above $T_c$[1,2]. This non-Fermi liquid behavior, often called the strange metal state, is characterized by a linearly increasing resistivity with temperature, counter to the quadratic behavior expected for electron-electron scattering in Landau's Fermi liquid (FL) theory (see for example ref. [3]). This linear-in-$T$ strange metal behavior is considered so unusual that it is believed by many to signal a new state of matter, motivating many of the most influential and exotic theoretical ideas of the cuprate problem including Anderson's resonating valence bond (RVB)[4], the marginal Fermi liquid (MFL)[5], many ideas about quantum critical points[6,7] as well as duals of string-theory models of quantum gravity[8]. Linearity in the measured resistivity has also recently been found in numerous other compounds, with connections to the cuprate physics sought[9]. Further, the strength of the coupling has often been described as the maximum possible, i.e. "Planckian"[10].

The strange metal state with linear-in-$T$ scattering occurs near the middle of the doping phase diagram, roughly above where the optimal or maximum $T_c$ exists. To the far right at high doping levels a regular (quadratic in $T$) Fermi liquid exists, while to the left at low doping levels is an unusual and poorly understood pseudogap state in which there is an incomplete suppression of low-energy spectral weight, especially at the antinodal regime of the Brillouin zone. And while tremendous effort has been invested to understand the pseudogap and strange metal states[11,12], a great amount of confusion still exists - for example, a recent optical spectroscopy study has argued that the normal state of the underdoped samples may obey a Fermi liquid quadratic-in-$\omega$ and quadratic-in-$T$ scaling of the scattering rates[13] —a result that would seem at odds with the general indications from the literature that the underdoped samples are more and more strongly correlated/less consistent with conventional Fermi Liquid physics.

Here we utilize angle resolved photoemission spectroscopy (ARPES) to study the electronic scattering rates or self-energies of the $Bi_2Sr_2CaCu_2O_8$ family of cuprate superconductors. The unique momentum-selectivity of ARPES allows it to measure the scattering rates in a simple and direct manner simply by looking at peak widths. We took advantage of the special ability of ARPES to measure the scattering rates as a function of both energy and temperature, whereas previous ARPES studies have relied on either the temperature or energy dependences alone. With this, we find a scaling in energy and temperature that we term power law liquid or PLL scaling, with the critical power law exponent varying smoothly with doping. The exponent is qualitatively consistent with expectations in the overdoped and optimally doped regimes, but diverges from the result of optics experiments in the underdoped regime [13]—something we discuss in detail. Further, we show that the coupling strength is effectively constant (when framed in the appropriate way) yet extremely strong throughout the entire doping range – a strength that is consistent with discussions of "Planckian" dissipation[10].

## Results and discussion

**Electronic interactions from ARPES**. Figure 1a presents ARPES data from an optimally doped $Bi_2Sr_2CaCu_2O_{8+\delta}$ sample taken in the normal state at $T = 100$ K. The data were taken along the nodal direction where the d-wave superconducting gap is zero. We use low energy (7 eV) photons, which give enhanced energy resolution, momentum resolution, and bulk sensitivity compared

to regular ARPES[14]. A slice through this spectrum at constant energy, known as a momentum distribution curve (MDC) is generally a Lorentzian whose width is $\Delta k_{MDC}(\omega)$, with this width directly proportional (through the electron velocity $dE/dk$—see Supplementary Note 2) to the single particle scattering rate, or equivalently, the imaginary part of the electron self-energy $\Sigma''(\omega)$. Through the Kramers–Kronig relation, its dispersion also directly connects to the real part of the electron self-energy $\Sigma'(\omega)$.

Figure 1b presents a compilation of the energy and temperature dependence of $\Sigma''(\omega)$ from five differently doped samples. As a function of energy each spectrum shows an approximately linear behavior at high energy with an upward curvature of the scattering rates near $E_F$ that are reminiscent of the Fermi liquid $\omega^2$ dependence, though as we will show this should not be taken as evidence of Fermi liquid behavior. As the temperature of any one sample is increased the curves shift up to higher scattering rates. Such a full set of ARPES scattering rate data as a function of energy and temperature (and doping) has not, to our knowledge, been previously presented.

**Form of electronic interactions**. Taking inspiration from (a) the success of the phenomenological Marginal Fermi Liquid form of self-energy proposed by Varma [5], and (b) the non-integral power laws in Anderson's Hidden Fermi liquid[15], we propose the following phenomenological Power Law Liquid or PLL form for the electronic scattering rates:

$$\Sigma''_{PLL}(\omega) = \Gamma_0 + \lambda \frac{\left[(\hbar\omega)^2 + (\beta k_B T)^2\right]^\alpha}{(\hbar\omega_N)^{2\alpha-1}} = \Gamma_0 + \lambda \frac{\left[(k_B T)^2\left[\left(\frac{\hbar\omega}{k_B T}\right)^2 + \beta^2\right]\right]^\alpha}{(\hbar\omega_N)^{2\alpha-1}}$$

(1)

where $\Sigma''(\omega)$ is the imaginary part of the self-energy directly measured in our experiment, $\Gamma_0$ is an offset parameter accounting for impurity or disorder scattering, $\lambda$ is a coupling parameter indicating the overall strength of the scattering, $\omega_N$ is a normalization frequency whose exponent maintains the proper dimensionality of the self-energy, parameter $\beta$ governs the comparative strengths of temperature and energy, and $\alpha$ is the critical PLL variable that takes the system from a FL to a MFL, and beyond. Note that this formalism does not directly contain any low energy scales that would be associated with superconductivity, the pseudogap, phonons, or other bosonic modes. The only energy scale is $\omega_N$ that is not strictly necessary, but we include to maintain constant units for coupling parameter $\lambda$ as a function of doping. Of all the energy scales in the system the one that closest matches our $\omega_N$ is the full conduction bandwidth relative to $E_F$. This form exhibits $\omega/T$ scaling only—a phenomenon commonly encountered in quantum critical types of theories.

The black dotted lines of Fig. 1b show fits of the data to the PLL self-energy, which reproduce the data extremely well up to the rather large scale of 0.1 eV scale that is considered here. This large energy range does cover the scale of potential bosonic modes (phonons, magnetic resonance), and while these have a noticeable effect on the spectra deep in the superconducting state they appear to only exist (magnetic resonance) or couple strongly (phonons) at low temperatures, i.e. in the superconducting or pairing regime. For example, both the very weak 10 meV kink[16] and the clear and very strong 50 meV kink that is dominant in the antinodal regime are only present below the onset of pairing[17]. The 70 meV-scale kink is also much weaker above $T_c$ than below $T_c$ [16], with more of a rounded effect compared to the sharper (in energy and k) kink effects below $T_c$, though the general energy scales of the centroids remain similar. Because of this, we believe it is likely that there are multiple contributions to the nodal kink —(a) the boson effect that is observed below $T_c$, and which might

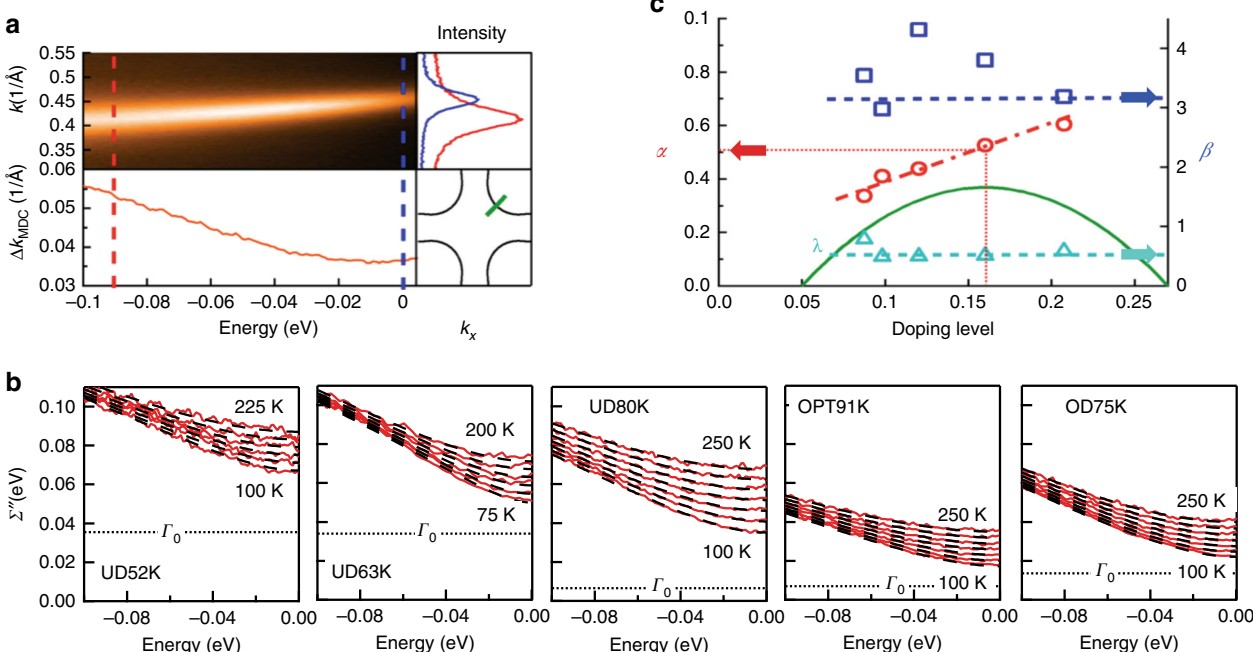

**Fig. 1** Experimental electronic self energy $\Sigma''$ as a function of energy, temperature, and doping. **a** The self energy is extracted from an ARPES spectrum (top left) located at the node (green cut, bottom right) by taking momentum cuts at constant energy (top right) to extract momentum widths $\Delta k_{MDC}(\omega)$ (bottom left) which are directly proportional to $\Sigma''(\omega)$. **b** Measured temperature and energy dependence of $\Sigma''$ for five different samples from underdoped $T_c = 52K$ (UD52K) through optimal doped $T_c = 91K$ (OPT91K) to overdoped $T_c = 75K$ (OD75K). **c** Fit results for the three main parameters in the model as a function of doping. The superconducting dome is schematically illustrated by the inverted parabola. The most relevant parameter is the power α which is seen to have a simple linear dependence on doping with value 0.5 very near optimal doping.

be due to coupling to phonons or the magnetic resonance mode, and (b) a smooth kink-like effect observed in the normal state due to a Kramers–Kronig transformation of the PLL scattering rate, as is shown in the Supplementary Fig. 2.

All curves for one sample have been fit simultaneously to Eq. 1, greatly constraining the parameter set that can fit the data. The extracted parameters as a function of doping are shown in Fig. 1c. One set of parameters $\alpha, \beta, \lambda$, and $\Gamma_0$ is obtained per sample, with these four parameters fitting all the ARPES data for all energies and all normal state temperatures. We do not include the normalization frequency $\omega_N$ here because it has been fixed at the high energy of 0.5 eV (approximately the bottom of the band) for all samples, which is purposely far beyond the 0.1 eV energy scale over which the data are fit, minimizing its impact on the obtained physics. $\omega_N$ is also fully mathematically irrelevant for the case that the parameter $\alpha = 1/2$ and almost irrelevant when $\alpha$ is near 1/2, as is the case for most important doping values.

Figure 1c shows that $\lambda$, and $\beta$ are essentially independent of doping level (and also energy and temperature). The $\beta$ values $(3.5 \pm 0.5)$ are close to the theoretical expectation of $\pi$ for a Fermi liquid metal (blue dashed line in Fig. 1c), with this result based upon the conversion of the Matsubara frequencies from the imaginary to real axes[18]. The main experiments that have addressed this issue in the literature are optics experiments, and then even for the simple Fermi liquid case, these have not successfully found the expected scaling between $T$ and $\omega$ (see ref. [13,19,20]). Therefore, the $\beta$ values uncovered here serve as a combination of a classic theoretical prediction, constraints on new theories, and a confirmation of the reasonableness of the PLL form of interactions. The offset parameter $\Gamma_0$ ranges from about 8–35 meV as a function of doping, as shown by the offset lines in panel b and discussed more extensively in Supplementary Note 5.

Parameter $\lambda$ is essentially constant as a function of doping, pegged to the value 0.5, again confirming the simplicity and

universality of the form of interactions. This value is roughly consistent with the recent notions of Planckian dissipation that stated that the optimally doped cuprates had the maximum possible scattering rate, determined by Planck's constant[10]. The fact that $\lambda$ is constant across the phase diagram implies that all doping levels may have this same maximum electronic coupling, though the form of the coupling (controlled by $\alpha$) varies strongly as a function of doping.

The data are shown in Fig. 1b is from 5 samples, 27 individual cuts of data, each of which contains on order of 50 energy points and 100 k points, or over $10^5$ data points total. For each sample, characterizing all of its data with the 5 parameters of Eq. 1 is impressive. That four of them essentially drop out leaving just the one linearly varying parameter $\alpha$ is even more so.

Figure 2 shows the imaginary self-energy (with $\Gamma_0$ subtracted off) for different doped samples vs. $\zeta^2 = \{(\hbar\omega)^2 + (\beta k_B T)^2\}$. Each plot contains many individual temperature curves that all collapse onto single lines, indicating the nearly ideal scaling behavior of the data as a function of temperature and frequency. On this log–log plot a power law function is perfectly linear, with the slope of the lines giving the Power Law Liquid exponent $\alpha$, as shown by the linear dashed lines on each of the plots, with these $\alpha$ values (slopes) gradually increasing from left to right.

The doping dependence of $\alpha$ is shown in more detail in Fig. 1c, indicating that it takes on a roughly linear dependence and is very close to 1/2 at optimal doping. In this case, Eq. 1 reduces to the hyperbolic form:

$$\Sigma''_{Opt} = \Gamma_0 + \lambda \sqrt{(\hbar\omega)^2 + (\beta k_B T)^2} \qquad (2)$$

which is linear in both energy (for $T = 0$) and temperature (for $\omega = 0$), i.e. it is of the MFL type of interaction [5] (see the theoretical plot of Fig. 3b) and the parameter $\omega_N$ also becomes irrelevant. If we extrapolate $\alpha$ to very high doping levels (HOD or

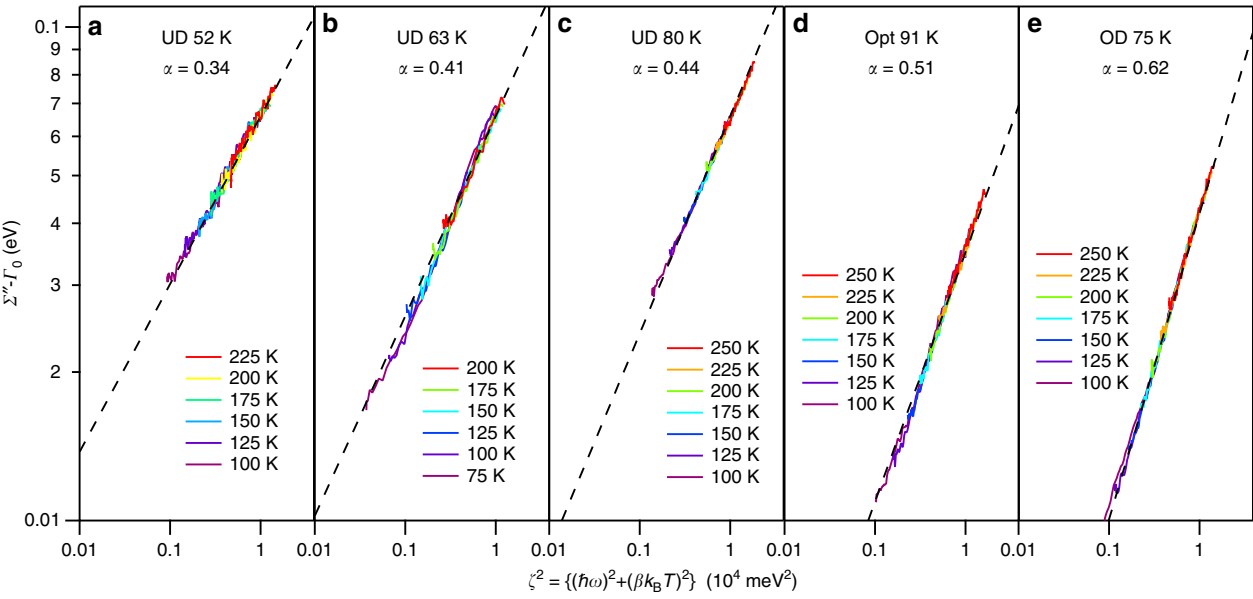

**Fig. 2** Imaginary self energy (with $\Gamma_0$ subtracted off) for different doped samples vs. $\zeta^2 = \{(\hbar\omega)^2 + (\beta k_B T)^2\}$. **a–e** Imaginary self-energy curves $\Sigma'' - \Gamma_0$ at multiple temperatures for 5 different samples, all collapse onto single lines with different power law component α. On this log–log plot a power law function is perfectly linear, with the slope of the lines giving the Power Law Liquid exponent α. β values for each sample were taken from Fig. 1b, i.e. were ~π.

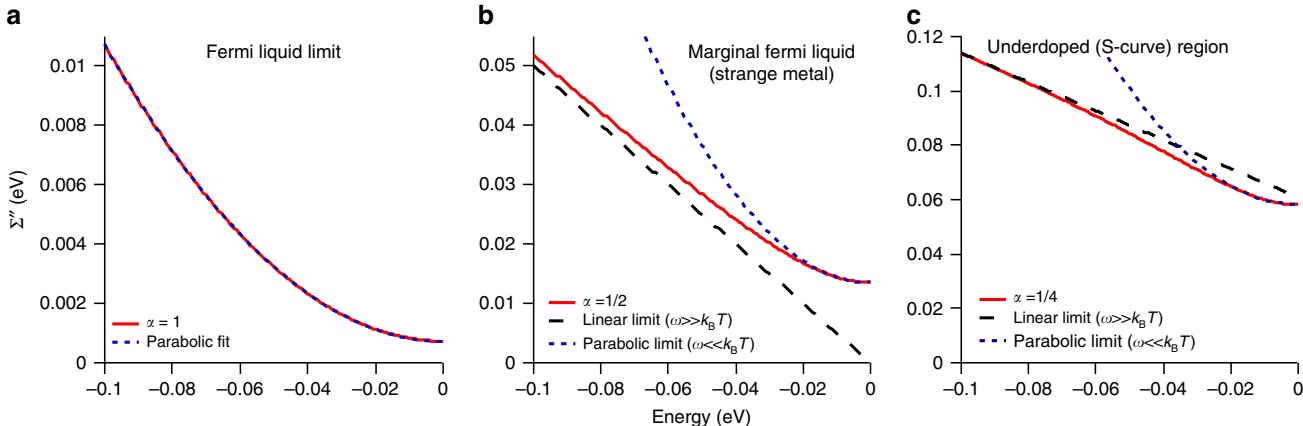

**Fig. 3** Theoretical plots of the imaginary part of the self energy for different doping regimes (red curves) for $T = 100$ K, $\beta = \pi$, $\lambda = 0.5$ and $\Gamma_0 = 0$. **a** In the extreme heavily overdoped (HOD) case ($\alpha = 1$) the system behaves as a normal Fermi liquid with a $\omega^2$ dependence. **b** In the optimally-doped case ($\alpha = 0.5$) the system can be approximated as having a $\omega^2$ dependence at low $\omega$ and linear at high $\omega$. **c** In the underdoped case ($\alpha = 0.25$) the system has an unusual s-curve $\omega$ dependence that shows a concave-up $\omega^2$ dependence at low energies and a concave-down form at higher energy.

Heavily Over Doped) such that $\alpha = 1$, the PLL form becomes:

$$\Sigma''_{\text{HOD}} = \Gamma_0 + \lambda \frac{(\hbar\omega)^2 + (\beta k_B T)^2}{\omega_N} \qquad (3)$$

and the quadratic dependence on energy and temperature of the normal Fermi liquid is recovered (Fig. 3a). As we go to the underdoped region the self-energy takes on an unusual S-curve shape with energy (Fig. 3c) that has the expected sub-linear behavior at higher energies. The behavior is a natural aspect of the PLL self-energy and is not possible with a linear combination of quadratic Fermi liquid and linear MFL which is necessarily concave up, in contrast to the underdoped data that is concave down at higher frequencies. To emphasize the S-curve in the underdoped regime we draw in the linear extrapolation of the deep energy dependence in Fig. 3c at a finite temperature of 100 K.

All curves in Fig. 3 are plotted at $T = 100$ K, and in all cases we show that the low energy behavior of the self-energies can be approximated by a parabolic curve (blue dashed lines), which would often be interpreted as indicative of the Fermi liquid $\omega^2$ scaling. It is therefore important that in the present case only Fig. 1a is an actual Fermi liquid, while Fig. 1c is a very special situation that (as we show later) even has zero quasiparticle residue at $T = 0$, i.e. it has zero overlap with Fermi liquid physics. Surprisingly, this form still has a quadratic-like behavior at low energies as shown in Fig. 3c, reminiscent of (but different than) a Fermi liquid.

**Temperature dependence and comparison to resistivity.** Figure 4a shows the calculated normal state temperature dependence of the $\omega = 0$ self-energy, using α values according to the linear fit (red dashed line) of Fig. 1c. We also fixed λ to 0.5 and β to π, and ignored the impurity scattering term $\Gamma_0$ because, as

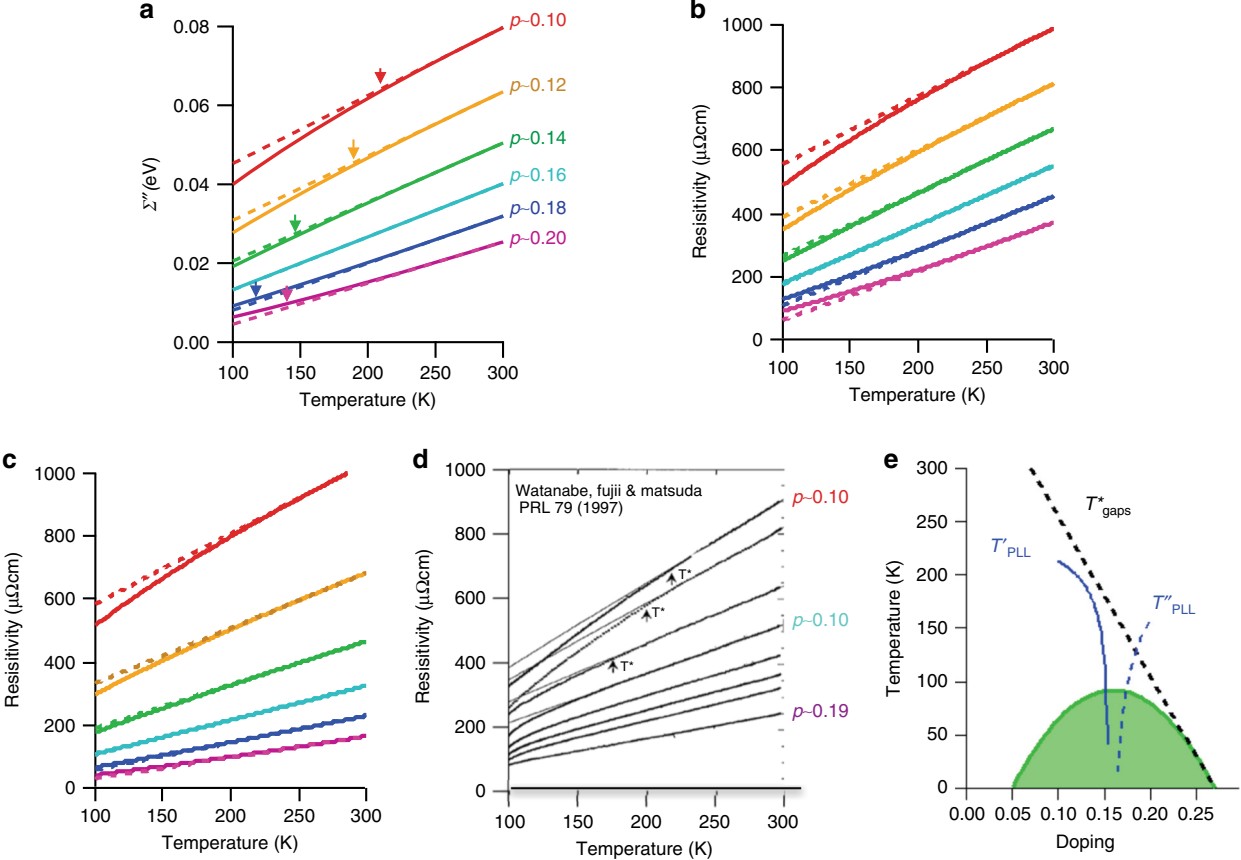

**Fig. 4** Scattering rates, resistivity, and apparent temperature scales. **a** Calculated doping and temperature dependence of the nodal $\Sigma''(\omega)$ (solid line) in the $\omega = 0$ limit using the linear relation between $\alpha$ and doping level $p$. Low temperature extrapolations from linear fits between [250 K, 300 K] are extrapolated as dashed lines. Additionally, $\Gamma_0 = 0$, $\beta = \pi$, and $\lambda = 0.5$ for all curves. **b, c** Resistivity as a function of temperature and doping calculated using two different methods (solid lines, see text for details). Linear extrapolations from [250 K, 300 K] are shown as dashed lines. The temperature dependence is dominated by the temperature dependence of $\Sigma''$. **d** Resistivity measurements from Watanabe et al. as well as the "pseudogap" temperature scale $T^\star$ [22]. **e** Compilation of the "break" temperatures from panel a: $T'$ and $T''$ are temperatures where there is an apparent break in $\Sigma''$ from the more linear form that is observed at high temperatures (up to 300 K).

discussed in Supplementary Note 5, this term is mostly from forward scattering contributions and/or chemical potential inhomogeneity that have minimal effects on the measured resistivity. Therefore there is essentially only one parameter $\alpha$ that created the entire set of curves shown in Fig. 4a. These curves have a roughly linear form at higher temperatures, with a deviation from linearity at lower temperature. The approximate temperature at which this deviation occurs is indicated by the arrows (see Supplementary Note 7), although this is not a sharply defined temperature scale.

With this information about the self-energy it is possible, within a few simple models, to calculate the temperature-dependent electrical resistivity. We do this in two standard but simplistic ways, shown for the Boltzmann transport model in Fig. 4b and the Drude model in Fig. 4c. The difference in these models comes largely from the way variations in the properties around the Fermi surface are considered, as discussed in Supplementary Note 6.

Figure 4d presents the measured resistivity of similar samples[21]. The overall scale and shape of the measured resistivity is surprisingly similar to that calculated from the self-energies (Figs. 4b, c). To our knowledge this is the closest agreement yet between the results of a transport measurement and a high-energy spectroscopy such as ARPES, also putting strong constraints on the origin of the strange metal resistive fluctuations. This agreement indicates that the electrical resistivity of the

cuprates can be closely connected to the single-particle electronic relaxation rates, with these relaxation channels dominated by large-angle scattering (since forward scattering contributes very weakly to the resistivity—see Supplementary Note 5). This would appear to make it difficult for theories with dominant coupling to $q \sim 0$ fluctuations [22–24] to connect to our data.

Conventionally, the temperature scale at which the resistivity deviates from the high temperature linear regime has been noted as $T^\star$ and has been one of the major tools used to determine the onset of the pseudogap phase. Using the same methods to extract this scale as used in transport, (Supplementary Note 7) a similar temperature scale vs. doping can be extracted from the PLL self-energies, as shown in Fig. 4a and summarized in Fig. 4e, but called $T^\star$ here for clarity. Note here that we have followed these temperature scales through the superconducting dome as if the superconductivity didn't disrupt the PLL phenomenology. Also plotted in Fig. 4e is an extracted temperature $T''$ in the overdoped regime, where the curvature turns upwards at low temperatures and hence also deviates from linearity. While the experimental transport data of Fig. 4d don't go far enough into the overdoped regime for the $T''$ scale to become visible, overdoped data from other families of the cuprates clearly show this power law behavior[25,26].

The extracted temperatures shown in Fig. 4e produce a v-shaped structure, with the crossover between the two branches (where there is perfect linearity) reaching $T = 0$ near optimal doping. This

v-shaped fan extending to zero temperature is reminiscent of the quantum critical behavior that has been extensively discussed in the cuprates [6], though the entire PLL phenomenology should perhaps better described as a quantum critical phase—that is, there is a very wide range of non-Fermi liquid phenomenology at zero temperatures, rather than only one doping level having this type of behavior. Despite that, the $T^*$ and $T''$ values as a function of doping look like the $T^*$ from other spectroscopies (Supplementary Fig. 1b). However, in contrast to the nodal scattering rate measurements discussed here, measurements of the spectral weight pseudogap from the gapping of energy spectra in the antinodal regime (also from ARPES on $Bi_2Sr_2CaCu_2O_8$) find a $T^*$ that asymptotically approaches the superconducting dome, such as shown in Supplementary Fig. 1a [12]. While the two different views of the phase diagram could previously have been justified as coming from different samples or types of spectroscopies, we now see that ARPES on BSCCO can give both types of phase diagrams, and that they, therefore, are both correct but are just measuring different phenomena. This, therefore, breaks the link between the temperature scales observed in transport (via scattering rates) and the temperature onset of a spectral weight pseudogap in underdoped samples (both of which have historically been called $T^*$).

**Quasiparticle residue and comparison to optics.** With the normal state nodal spectral function as written down in the PLL form we can determine many properties in addition to resistivity, with two of them described here. The quasiparticle residue $Z$ is a concept from Landau's Fermi liquid theory telling the quasiparticle weight, which in a true Fermi liquid must be finite but potentially very small (like in a heavy fermion). Again assuming we can kill the superconductivity so as to stay in the PLL phase all the way to zero temperature, we calculate $Z$ (Supplementary Note 8) with results shown in Fig. 5a. The PLL $Z$ is finite for all overdoped samples and exactly at $T=0$ and $\omega=0$ it is identically zero (a true non-Fermi liquid) for all underdoped samples. The transition between them occurs at $\alpha=1/2$ (i.e., the marginal Fermi liquid case) and $T=0$ and could be considered a quantum critical point, possibly connecting to the v-shaped form of Fig. 4e that is also reminiscent of quantum criticality.

Despite the unusual power laws in the scattering rates, the PLL spectral weight shows a regular (non-power-law) metallic Fermi edge in all cases consistent with the ARPES data. This is different from other models in which power law scattering rates have been proposed, such as the Projected Fermi Liquid of Anderson[27] and

the famous Luttinger Liquid that is known to exist in one dimension[28]. Despite the zero quasiparticle residue at $T=0$, when observed at finite temperatures a quadratic $\omega^2$ dependence to the scattering rate can still be found at low frequency near $E_F$ on the underdoped side (Fig. 3c). Taken together, this phenomenology warns us that the traditional view of quasiparticle vs non-quasiparticle (looking for well-defined Fermi edge, $\omega^2$ dependence, etc.) might be too simplistic.

Using PLL self-energies, we can also simulate the optical conductivity (Supplementary Note 9) as a function of doping, as shown in Fig. 5b for a simulation temperature $T=300$ K, compared to measured optical data (Fig. 5c) at the same temperature/dopings[29]. The overall agreement is highly satisfactory, including both the magnitude of the conductivity and the width of the low energy peaks. In general, there has been a great deal of debate about the nature of the low energy spectral peaks, especially whether these should be considered Drude peaks representative of a quasiparticle-based Fermi liquid. Here we see that even though in the strictest sense there are no quasiparticle peaks for $\alpha < 0.5$ (i.e. the zero temperature quasiparticle residues vanish), there are zero-frequency upturns in the optical simulations and data that look something like a Drude peak.

**Possible origins of PLL phenomenology.** Just as the origin of the MFL phenomenology is still not known, we expect the origin of the PLL phenomenology will take time to sort out. However, we hope that the extra constraints of this phenomenology compared to that of the MFL phenomenology will give a boost to our theoretical understanding of these materials.

Strange as it is, the power law form of self-energy is not unprecedented but perhaps under-explored. Fractional power law forms of self-energy have been identified in iron-chalcogenide and ruthenate superconductors, and have been attributed to orbital and spin fluctuations[30]. The case of $\alpha=0.5$ might have been observed in scattering rates of pnictides in a magnetic field with $\mu_B\mathbf{B}$ in place of $\omega$, possibly related to the proximity of a quantum critical point[31]. The power law form has also been proposed as an outcome of the extended Hubbard model in infinite dimensions[32], or as the result of un-particles in the anti–de Sitter/conformal field theory (AdS/CFT) correspondence[33]. It also is present in the SYK (Sachdev-Ye-Kitaev) model of fully incoherent electrons[34,35], which nominally would correspond to our exponent $\alpha=1/4$, and that has recently been shown to be able to support coherent Cooper pairing[36]. Ref. 36

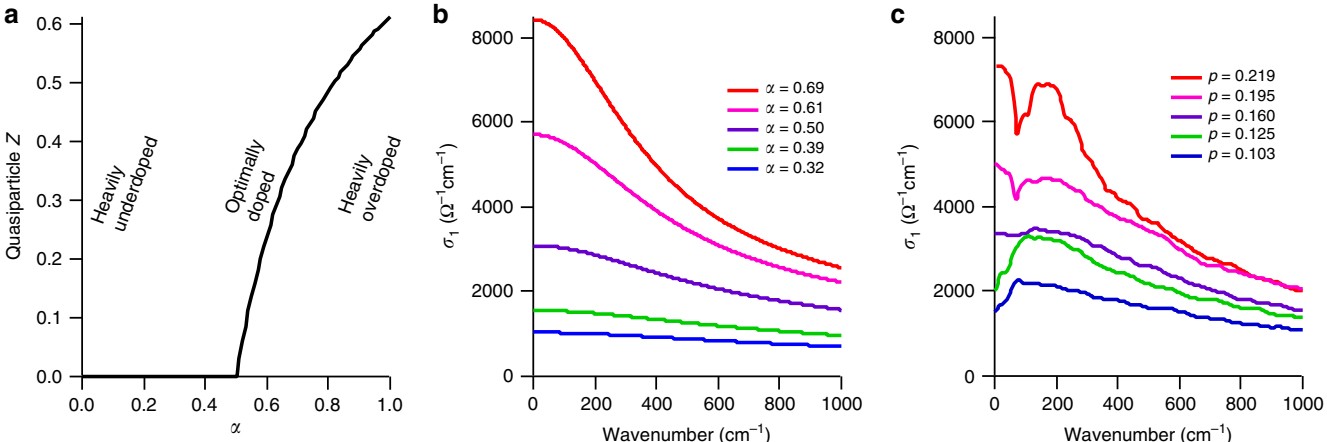

**Fig. 5** Quasiparticle residue and optical conductivity. **a** The quasiparticle residue $Z$ calculated using our PLL functional form (also see Supplementary Note 8). **b** Simulated and **c** measured normal state ($T=300$ K) optical conductivity (reconstructed from ref. [30], see Supplementary Note 9) as a function of doping. In **b** $\alpha$ values were chosen to match the reported doping levels in **c**.

also shows that changing the ratio m of the number of boson and fermion flavors allows the power law exponent to change continuously over a ratio of 2, which could have connections to the running exponent we observe in our data. Whether the universal power law form in cuprates and the similar forms in other materials can be unified under a single theoretical framework is an intriguing question calling upon further theoretical and experimental explorations.

A widely discussed explanation for the linear MFL behavior near optimal doping is based upon fluctuations above a quantum critical point, in which two phases meet at zero temperature from opposite sides of the phase diagram [6]. The v-shaped quantum-critical-like behavior shown in Fig. 4e as well as in Supplementary Fig. 1b seems to support this possibility, though here we note that our results are inconsistent with the underdoped regime being Fermi-liquid-like. Rather, our results indicate that huge doping regimes are $Z = 0$ non-Fermi-liquids and so we may be better off discussing these materials in the context of a set of quantum critical phases[37] rather than as a single quantum critical point.

**Outlook**. While the PLL ansatz is not necessarily a unique choice for a self-energy, it has a remarkable simplicity that with one smoothly varying parameter as a function of doping (the power law exponent $\alpha$), we can understand the salient normal-state features of the strange-metal ARPES, transport, and optics data. Understanding the origin of this power law behavior therefore holds great promise for understanding the cuprates—not just for the strange metal normal state behavior but also for the superconducting state since it is born out of this power-law strange-metal state.

## Data availability

The datasets used in the current study are available from the corresponding author on reasonable request.

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

## Acknowledgements

We thank A. Chubukov, M. Hermele, L. Radzihovsky, and J. Schmalian for valuable conversations and D.H. Lu and R.G. Moore for help at SSRL. SSRL is operated by the DOE, Office of Basic Energy Sciences. Funding for this research was provided by DOE Grant No. DE-FG02-03ER46066 (Colorado) and DE-AC02-98CH10886 (Brookhaven).

## Author contributions

T.J.R. and D.S.D. conceived the experiment. T.J.R. carried out ARPES measurements with the help of N.C.P., S.P., J.A.W., Y.C., Z.S., H.L., Q.W. and X.Q.Z. J.S.W., Z.J.X., G.G., Y.Y., and H.E. grew the high quality crystals. T.J.R. and X.Q.Z. analyzed the data. T.J.R., X.Q.Z., H.L., G.A. and D.S.D. discussed the results. T.J.R., X.Q.Z. and D.S.D. wrote the paper.

## Competing interests

The authors declare no competing interests.
