## [Peer Review File · Nature Communications]

I have gone through the manuscript by T. J. Reber *et al.* In this manuscript, the authors present detailed self-energy analysis of ARPES data from BISCO 2212 cuprate high temperature superconductor (HTSC) samples as a function of temperature and carrier concentration. Based on this, they claim that a unique functional form of the single-particle self-energy underlies the normal state phase diagram of this material. If this can be categorically proved, it will possibly have major implications to the theoretical models of cuprate HTSCs. In short, the topic of the manuscript is important, and can be of interest to both theorists and experimentalists in the field of HTSCs and possibly, other strongly correlated electron systems.

However, I feel that the manuscript has various issues as listed below. In particular, I have a number of concerns and questions regarding the analysis and presentation of the ARPES data in the manuscript. These are to be duly addressed before the manuscript can be considered for publication in Nature Communications.

(A) Results and Conclusions:

(i) Question regarding the imaginary (Σ'') part of the single-particle self-energy data presented in Fig. 1:

In Fig. 1, the authors have presented the data on $\Sigma''(\omega)$ for several doping values. In addition to the kink at $\omega \sim 70$ meV, previous LASER ARPES measurements have reported the so-called low-energy kink at $\omega \sim 10$ -15 meV for BISCO 2212. I can understand that the signature of the kink at ~ 70 meV may not be easily visible in the data because of the limited ω -range. But, one would expect that at least the low energy kink structure should be observable in $\Sigma''(\omega)$ in Fig. 1. This doesn't seem to be the case—at least from a quick glance of Fig. 1. The authors should elaborate on this.

(ii) How robust is the functional form of $\Sigma''(\omega)$?

The central observation of the paper is the form of the self-energy as a function ω and T described by Equation 1. For a given value of T, Equation 1 reduces to Equations 2 and 3 for OPT/UD and OD samples, respectively. One of the main conclusions of the manuscript is the crossover from Fermi liquid to non-Fermi liquid behavior of the self-energy with a decrease in doping from the overdoped to the underdoped regime of the phase diagram. This hinges on the authors' claim that "a possible combination of quadratic Fermi liquid and linear MFL" can't fit $\Sigma''(\omega)$ of an underdoped sample. I suggest that the authors present $\Sigma''(\omega)$ data points together with the fitted curves in Fig. 2 and establish that this is indeed the case.

I would like to bring the following paper to the attention of the authors: "*Spectroscopic evidence for Fermi liquid-like energy and temperature dependence of the relaxation rate in the pseudogap phase of the cuprates*" (PNAS, **110**, 5774–5778 (2013)), which reports Fermi liquid behavior of the scattering rate (based on optics data from a few *different types* of cuprate HTSCs) even in the underdoped regime. This paper reports that $1/\tau(\omega, T)$ follows a functional form that is expected for a Fermi liquid system, where $\tau(\omega, T)$ is the electronic lifetime. In other words, the results of this paper are qualitatively different from those of the current manuscript. Given the success of the Drude model in terms of capturing electrical transport in BISCO2212 (as reported by the

authors in this manuscript), I feel that the results of this paper in PNAS is relevant to the current manuscript. One can raise various questions: Is BISCO 2212 a special case where such power law liquid form of the self-energy is applicable? Could it be that the scatterings from intrinsic disorders of BISCO 2212 somehow mask the Fermi liquid behavior in BISCO 2212? In fact, such a possibility has been suggested in the context of BISCO 2201 in the PNAS paper.

(iii) Zero quasiparticle residue despite Fermi liquid behavior in the vicinity of $\omega \sim 0$:

From Fig. 2, the authors claim that a Fermi Liquid behavior is present at all doping levels in the low-energy regime. It is somewhat difficult to reconcile this with the other observation of the manuscript, namely zero quasiparticle residue (at $\omega=0$) below optimal doping. I would like the authors to comment on this.

(iv) Presentation of the real part of the self-energy $\Sigma'(\omega)$ data:

In Fig. 4a, the authors discussed the doping evolution of quasiparticle residue Z at the nodal point. Quite remarkably, the authors find that Z goes to identically zero for some doping level just below the optimal one. As to my understanding, this observation is quite different from the conventional wisdom in the field—even though the antinodal quasiparticles in BISCO 2212 HTSCs disappear at lower doping values, the nodal quasiparticles remain intact. Nevertheless, I am intrigued by the possible consequences of the above result. As described in the supplementary section, Z was calculated using: $1/[1-\{d\Sigma(\omega)/d\omega\}_{\omega=0}]$, assuming ω is referenced with respect to the chemical potential. The renormalization of the electronic dispersion is a measure of the mass renormalization. Therefore, Z and the mass renormalization are related to each other. More specifically, Z is inversely proportional to the mass renormalization. Consequently, effective mass should diverge at doping values for which $Z=0$. Is it really the case? The authors need to show the $\Sigma'(\omega)$ data for different doping levels similarly to the $\Sigma''(\omega)$ data in Fig. 1(c).

(v) Data from other points on the Fermi surface:

Here, the entire phase diagram has been characterized using data only at the node. As I realized by reading the manuscript, the authors have extensive data set from other points on the Fermi surface as well. I strongly encourage the authors to include such data in the manuscript.

(B) Minor issue regarding rewriting of the abstract:

The authors should rewrite the abstract of the manuscript such that its novelty is easily accessible to the general readers. Currently, the abstract seems to be geared very much towards the experts in the field.

In summary, even though it deals with an important topic, the manuscript in its current form is not suitable for publication in Nature Communications. However, it can be considered for publication provided the authors carefully address the above-described issues.

Reviewer #2 (Remarks to the Author):

The manuscript reports the phenomenological fitting formula for the imaginary part of the self-energy, which the authors use to explore the ARPES, optical conductivity and resistivity data.

1) First let me state that from a formal theory perspective the fitting procedure with running exponent is not well justified to my opinion. Furthermore, it is well known that the leading T dependence of $\text{Im } \Sigma$ comes from the thermal piece, which does not fit into ω/T scaling.

Nevertheless, if one forgets about the thermal piece and just consider $T=0$ and analyze how $\text{Im } \Sigma(\omega)$ changes as a system moves away from a quantum critical point, one expects to get $\text{Im } \Sigma \propto \omega^2$ in a progressively wider scale, and then some ω^a with $a < 1$. In this regard the proposal to use a phenomenological ansatz $\omega^{[2b]}$ allows a "somewhat" reasonable fit with b increasing away from a QCP. This is what the authors do.

In this regard, I find the present manuscript deserves publication in Nature Communication as a careful experimental analysis of a proximity to the quantum critical point. However, I request the authors to make careful remarks in the manuscript on the purely phenomenological nature of this fit and the temperature component.

2) I wonder how the authors excluded the coupling to the bosonic modes which are also present along the nodal direction and if yes whether they could subtract these effects.

Report of the first referee and authors' reply:

Referee: I have gone through the manuscript by T. J. Reber *et al.* In this manuscript, the authors present detailed self-energy analysis of ARPES data from BISCO 2212 cuprate high temperature superconductor (HTSC) samples as a function of temperature and carrier concentration. Based on this, they claim that a unique functional form of the single-particle self-energy underlies the normal state phase diagram of this material. If this can be categorically proved, it will possibly have major implications to the theoretical models of cuprate HTSCs. In short, the topic of the manuscript is important, and can be of interest to both theorists and experimentalists in the field of HTSCs and possibly, other strongly correlated electron systems.

However, I feel that the manuscript has various issues as listed below. In particular, I have a number of concerns and questions regarding the analysis and presentation of the ARPES data in the manuscript. These are to be duly addressed before the manuscript can be considered for publication in Nature Communications.

Reply:

We thank the referee for the positive comments, and answer the referee's concerns and questions point-by-point below.

Referee:

(A) Results and Conclusions:

(i) *Question regarding the imaginary (Σ'') part of the single-particle self-energy data presented in Fig. 1:*

In Fig. 1, the authors have presented the data on $\Sigma''(\omega)$ for several doping values. In addition to the kink at $\omega \sim 70$ meV, previous LASER ARPES measurements have reported the so-called low energy kink at $\omega \sim 10$ -15 meV for BISCO 2212. I can understand that the signature of the kink at ~ 70 meV may not be easily visible in the data because of the limited ω -range. But, one would expect that at least the low energy kink structure should be observable in $\Sigma''(\omega)$ in Fig. 1. This doesn't seem to be the case—at least from a quick glance of Fig. 1. The authors should elaborate on this.

Reply:

We thank the referee for pointing out this concern. All of the kinks observed in the low energy dispersion and scattering rates in the cuprates are very strongly temperature dependent, with most of them only observable at low temperatures for $T < T_c$ (or $T < T_{\text{pair}}$). The 10 meV kink discussed above is one key example of this [**Phys. Rev. Lett.** 105, 046402 (2010)]. Another example is the 50 meV kink in the antinodal regime, which is the strongest and clearest kink and therefore the most carefully studied. It is also found that this kink is only present below T_c [**Nat. Comm.** 9, 36 (2018)]. Finally, there is the 70 meV-scale kink

observed along the nodal direction, which is weaker than the 50 meV kink discussed above. This kink is also much weaker above T_c than below T_c [**Phys. Rev. Lett.** 105, 046402 (2010)], with more of a rounded effect compared to the sharper (in energy and k) kink effects below T_c , though the general energy scales of the centroids remain similar. Because of this, we believe it is plausible that there are multiple contributions to the nodal kink – a) the boson effect that is observed below T_c , b) a smooth kink-like effect observed in the normal state due to a Kramers-Kronig transformation of the PLL scattering rate, as is shown in the figure 2 below. We have discussed this in more detail in the resubmitted version of the manuscript.

Referee:

(ii) *How robust is the functional form of $\Sigma''(\omega)$?*

The central observation of the paper is the form of the self-energy as a function ω and T described by Equation 1. For a given value of T , Equation 1 reduces to Equations 2 and 3 for OPT/UD and OD samples, respectively. One of the main conclusions of the manuscript is the crossover from Fermi liquid to non-Fermi liquid behavior of the self-energy with a decrease in doping from the overdoped to the underdoped regime of the phase diagram. This hinges on the authors’ claim that “a possible combination of quadratic Fermi liquid and linear MFL” can’t fit $\Sigma''(\omega)$ of an underdoped sample. I suggest that the authors present $\Sigma''(\omega)$ data points together with the fitted curves in Fig. 2 and establish that this is indeed the case.

Reply:

As shown in Fig. 2c, on the underdoped side the imaginary self energy has an “S” shape, i.e. it contains both a concave up portion (near E_F) and a concave down portion farther away from E_F . In contrast, any combination of regular Fermi liquid form (parabolic) and a marginal Fermi liquid form (linear) at a given temperature is necessarily concave up, as it can be generally written as a second order polynomial function of ω . Our power law liquid form on the other hand fit the data much better (a reduced Chi-squared over three times better for the PLL fit than for the polynomial fit). This is demonstrated by the example below:

Figure 1. a) Example of the underdoped sample data ($T_c=63$ K) fit to linear and parabolic form. b) Example of the underdoped sample data ($T_c=63$ K) fit to the power law liquid (PLL)

form, compared with *c*) fitting to a combination of linear and parabolic form, i.e. a second order polynomial function. The variance of the PLL fit is 3 times smaller than the polynomial fit.

Per referee's suggestion, we added this discussion into our supplementary information.

Referee:

I would like to bring the following paper to the attention of the authors: “*Spectroscopic evidence for Fermi liquid-like energy and temperature dependence of the relaxation rate in the pseudogap phase of the cuprates*” (PNAS, **110**, 5774–5778 (2013)), which reports Fermi liquid behavior of the scattering rate (based on optics data from a few *different types* of cuprate HTSCs) even in the underdoped regime. This paper reports that $1/\tau(\omega, T)$ follows a functional form that is expected for a Fermi liquid system, where $\tau(\omega, T)$ is the electronic lifetime. In other words, the results of this paper are qualitatively different from those of the current manuscript. Given the success of the Drude model in terms of capturing electrical transport in BISCO2212 (as reported by the authors in this manuscript), I feel that the results of this paper in PNAS is relevant to the current manuscript. One can raise various questions: Is BISCO 2212 a special case where such power law liquid form of the self-energy is applicable? Could it be that the scatterings from intrinsic disorders of BISCO 2212 somehow mask the Fermi liquid behavior in BISCO 2212? In fact, such a possibility has been suggested in the context of BISCO 2201 in the PNAS paper.

Reply:

We thank the referee for mentioning this paper, which we had discussed in the previous version of our Supplementary Materials but not in the main text of the paper. Since we agree that it is an important part of the literature we have extended our discussion on this topic and moved some of it to the main part of the paper. We discuss the main aspects here:

- We show that that their optical data actually fits better to the PLL form than to the Fermi Liquid form, extracting Power Law Liquid exponents α between 0.52 and 0.73 for their three samples, which contrast with the expectation of $\alpha=1$ for a Fermi Liquid. We show this new analysis in our supplementary materials Fig S6.
- If we compare the fits of our underdoped ARPES data using a PLL form vs. a Fermi Liquid form we find that the PLL form is visually much better than the fit to the Fermi Liquid form (Fig S7 Supplementary Materials) and has a variance from the fit results that is 6 times smaller than that from the Fermi liquid form.
- We note that other aspects of the Fermi liquid analysis of the optics data do not hold together as well as they do for the PLL analysis of the ARPES data, including:
 - o The prefactor connecting the ω^2 and T^2 portions (which they term $p*\pi$) returns a value $p=1.5$ for them, while $p=2$ is expected for optics. The prefactor that we determine (which we termed β) is exactly as expected for all doping levels, i.e. $\beta=\pi$.
 - o The offset values that are obtained from the optical fits are unusual, with some of them negative, which we feel to be unphysical. These were not stated in their paper but are the

results of our fits to their data, mentioned in the caption to supplementary Figure S5. Our offset values Γ_0 are shown for all of our samples in Fig 1b of our main paper and are all reasonable.

Referee:

(iii) *Zero quasiparticle residue despite Fermi liquid behavior in the vicinity of $\omega \sim 0$:*

From Fig. 2, the authors claim that a Fermi Liquid behavior is present at all doping levels in the low-energy regime. It is somewhat difficult to reconcile this with the other observation of the manuscript, namely zero quasiparticle residue (at $\omega=0$) below optimal doping. I would like the authors to comment on this.

Reply:

Even though the low energy portions of all the curves of figure 2 appear to be able to be fit with a quadratic in ω form we do not claim that it is FL behavior – rather we claim the opposite – panels 2b and 2c (optimal and underdoped) are not FLs and in fact do have zero residue in the limit of $\omega=0$ and $T=0$. What we show in the manuscript, is that the ω^2 form of scattering can appear to approximate the low energy scattering data when $k_B T > \omega$, but this should NOT be taken to imply Fermi Liquid scattering, especially because the quasiparticle residue does go to identically zero for all UD samples at $T=0$ (see answer to next question). Therefore, the generally-accepted view to only look for ω^2 at low frequencies is too simplistic. We have clarified this point in the resubmitted manuscript.

Referee:

(iv) *Presentation of the real part of the self-energy $\Sigma'(\omega)$ data:*

In Fig. 4a, the authors discussed the doping evolution of quasiparticle residue Z at the nodal point. Quite remarkably, the authors find that Z goes to identically zero for some doping level just below the optimal one. As to my understanding, this observation is quite different from the conventional wisdom in the field—even though the antinodal quasiparticles in BISCO 2212 HTSCs disappear at lower doping values, the nodal quasiparticles remain intact. Nevertheless, I am intrigued by the possible consequences of the above result. As described in the supplementary section, Z was calculated using: $1/[1 - \{\Sigma'(\omega)/d\omega\}_{\omega=0}]$, assuming ω is referenced with respect to the chemical potential. The renormalization of the electronic dispersion is a measure of the mass renormalization. Therefore, Z and the mass renormalization are related to each other. More specifically, Z is inversely proportional to the mass renormalization. Consequently, effective mass should diverge at doping values for which $Z=0$. Is it really the case? The authors need to show the $\Sigma'(\omega)$ data for different doping levels similarly to the $\Sigma''(\omega)$ data in Fig. 1(c).

Reply:

The referee is correct that the effective mass should (and does) diverge at the doping values for which $Z=0$, though in fact this divergence only occurs for the case where $\omega=0$ and $T=0$.

This is shown in the figure below, with the first two panels included as Fig S2 in the supplemental material. The other panels will be included in a separate manuscript that will discuss the implication of zero residue Z. Below is a figure from that manuscript in regards to the doping and temperature dependence of $\Sigma'(\omega)$, $d\Sigma'(\omega)/d\omega$ and Z:

Figure 2. Calculation of the imaginary part a) and real part b) of the power law liquid self-energy Σ at $T=1$ K for various parameters α . c) The energy derivative of the real part Σ' at zero is inversely proportional to the quasiparticle residue Z. d) The quasiparticle residue Z as a function of power law liquid parameter α at multiple temperatures.

Panels a,b, and c show Σ'' , Σ' , $\delta\Sigma'/\delta\omega$ ($\omega=0$), and Z for many alpha values, all at the low but finite temperature $T=1$ K. The Kramer's Kronig transformation to obtain Σ' from Σ'' was carried out with an energy cutoff of 0.5 eV, i.e. very far beyond the 0.1 eV energy scale shown in the figures, and we have confirmed that the results are only very weakly dependent on the cutoff energy. An important thing to note is that the slope of Σ' at E_F is increasingly steeper as α decreases (more underdoping), as shown in panel b. This is shown in red in panel c. In black in panel c is the residue $Z=1+|\delta\Sigma'/\delta\omega$ ($\omega=0$) that gets smaller and smaller as α decreases.

Panel d shows the black points of panel c in blue ($T=1\text{K}$) as well as for many other temperatures up to 200K (red) and for $T=0\text{K}$ (black). The $T=0\text{K}$ plot is the curve plotted in Fig 4a in the main paper. The trends are clear – rising temperature increases the quasiparticle Z , giving a finite residue for any non-zero temperature, while for $T=0$ (and $\omega=0$) the residue is identically zero for all $\alpha \leq 0.5$. Note that the $\alpha=0.5$ result is completely consistent with the famous “Marginal Fermi Liquid” result of Varma et al. [**Phys. Rev. Lett.** 63, 1996 (1989)], which has the same linear scattering rate as our $\alpha=0.5$ Power Law Liquid and which has been stated to have zero residue – but only “marginally”, i.e. it is right on the edge of having a residue and not having one – just as our simulations show. We have clarified this point in the resubmitted manuscript.

Referee:

(v) Data from other points on the Fermi surface:

Here, the entire phase diagram has been characterized using data only at the node. As I realized by reading the manuscript, the authors have extensive data set from other points on the Fermi surface as well. I strongly encourage the authors to include such data in the manuscript.

Reply:

The referee is certainly correct that we have made many measurements around the Fermi surface. However, away from the nodal direction, it becomes increasingly difficult to accurately extract $\Sigma''(\omega)$ and $\Sigma'(\omega)$ using the MDC (Momentum Distribution Curves) methods used in this paper. To overcome this, our group developed an alternative method, as shown in a separate manuscript [*Nat. Comm.* 9, 26 (2018)]. For any one sample the overall slope of $\Sigma''(\omega)$ seems to stay roughly the same as we go away from the node, but with a rising offset that is not yet well understood, but the analysis is more complicated, requires much more extensive data, and is beyond the scope of this work. In the present paper, we focused on the nodal direction as it is more closely related to the Marginal Fermi liquid and transport and optics experiments (which are dominated by the nodal states that have the largest velocities) that we can make direct connections to.

Referee:

(B) Minor issue regarding rewriting of the abstract:

The authors should rewrite the abstract of the manuscript such that its novelty is easily accessible to the general readers. Currently, the abstract seems to be geared very much towards the experts in the field.

Reply:

Per the referee’s suggestion, we have rewritten the abstract to be more suitable for the general audience.

Referee:

In summary, even though it deals with an important topic, the manuscript in its current form is not suitable for publication in Nature Communications. However, it can be considered for publication provided the authors carefully address the above-described issues.

Reply:

We believe that all concerns have now been fully addressed.

Report of the second referee and authors' reply

Referee:

The manuscript reports the phenomenological fitting formula for the imaginary part of the self-energy, which the authors use to explore the ARPES, optical conductivity and resistivity data.

1) First let me state that from a formal theory perspective the fitting procedure with running exponent is not well justified to my opinion. Furthermore, it is well can be that the leading T dependence of Σ comes from the thermal piece, which does not fit into ω/T scaling.

Reply:

The entire regime of incoherent non-Fermi liquid physics is in its infancy, and so it not fully justified – however it is an extremely important area of physics both experimentally and theoretically that many people feel is needed for understanding these materials. A recent theoretical work in this area is the celebrated SYK or Sachdev-Ye-Kitaev model (our new references 34,35) that describes fully incoherent electrons with a power law “ Δ ” equivalent to our $\alpha=1/4$ (our $\alpha=0.5-\Delta$ in their terminology). Esterlis and Schmalian (our ref 36) show (among other things) that changing the ratio m of the number of boson and fermion flavors allows the exponent Δ to change continuously over a ratio of 2. These references and discussion have been added to our revised manuscript.

Referee:

Nevertheless, if one forgets about the thermal piece and just consider $T=0$ and analyze how $\Sigma(\omega)$ changes as a system moves away from a quantum critical point, one expects to get $\text{Im} \Sigma \propto \omega^2$ in a progressively wider scale, and then some ω^a with a <1 . In this regard the proposal to use a phenomenological ansatz $\omega^{[2b]}$ allows a "somewhat" reasonable fit with b increasing away from a QCP. This is what the authors do.

In this regard, I find the present manuscript deserves publication in Nature Communication as a careful experimental analysis of a proximity to the quantum critical point.

However, I request the authors to make careful remarks in the manuscript on the purely phenomenological nature of this fit and the temperature component.

Reply:

We thank the referee for the positive comments, and make changes to the manuscript stressing that our fits are purely phenomenological.

Referee:

2) I wonder how the authors excluded the coupling to the bosonic modes which are also present along the nodal direction and if yes whether they could subtract these effects.

Reply:

This energy range analyzed in our paper does cover the scale of potential bosonic modes (phonons, magnetic resonance), and while these have a noticeable effect on the spectra deep in the superconducting state they appear to only exist (magnetic resonance) or couple strongly (phonons) at low temperatures, i.e. in the superconducting or pairing regime. For example, both the very weak 10 meV kink [**Phys. Rev. Lett.** 105, 046402 (2010)] and the clear and very strong 50 meV kink that is dominant in the antinodal regime are only present below the onset of pairing [**Nat. Comm.** 9, 26 (2018)]. The 70 meV-scale kink is also much weaker above T_c than below T_c [**Phys. Rev. Lett.** 105, 046402 (2010)], with more of a rounded effect compared to the sharper (in energy and k) kink effects below T_c , though the general energy scales of the centroids remain similar. Because of this, we believe it is likely that there are multiple contributions to the nodal kink – a) the boson effect that is observed below T_c , and which might be due to coupling to phonons or the magnetic resonance mode, and b) a smooth kink-like effect observed in the normal state due to a Kramers-Kronig transformation of the PLL scattering rate, as is shown in the new supplemental figure S2, included here. This implies that for the normal state data analyzed here a subtraction of the “boson effect” is likely either unnecessary or inappropriate.

All of this is now discussed more clearly in the present manuscript.

Figure S2. Calculation of the imaginary part a) and real part b) of the power law liquid self-energy Σ at $T=1$ K for various parameters a . Note that the curve in S' will cause a kink-like band dispersion in the energy range where the curvature is observed, i.e. 0-70 meV.

REVIEWERS' COMMENTS:

Reviewer #1 (Remarks to the Author):

The authors addressed the issues, I raised in my report. In my opinion, this manuscript can now be published.

Reviewer #2 (Remarks to the Author):

I believe the authors did their best to answer the comments of the referee and I recommend the publication